# MF-DCMANet: A Multi-Feature Dual-Stage Cross Manifold Attention Network for PolSAR Target Recognition

**Feng Li** [1,2]**, Chaoqi Zhang** [1]**, Xin Zhang** [1,2,*] **and Yang Li** [1,2]

1. Radar Research Laboratory, Beijing Institute of Technology, Beijing 100081, China
2. Beijing Institute of Technology Chongqing Innovation Center, Chongqing 401120, China
* Correspondence: xin.zhang@bit.edu.cn

**Abstract:** The distinctive polarization information of polarimetric SAR (PolSAR) has been widely applied to terrain classification but is rarely used for PolSAR target recognition. The target recognition strategies built upon multi-feature have gained favor among researchers due to their ability to provide diverse classification information. The paper introduces a robust multi-feature cross-fusion approach, i.e., a multi-feature dual-stage cross manifold attention network, namely, MF-DCMANet, which essentially relies on the complementary information between different features to enhance the representation ability of targets. In the first-stage process, a Cross-Feature-Network (CFN) module is proposed to mine the middle-level semantic information of monogenic features and polarization features extracted from the PolSAR target. In the second-stage process, a Cross-Manifold-Attention (CMA) transformer is proposed, which takes the input features represented on the Grassmann manifold to mine the nonlinear relationship between features so that rich and fine-grained features can be captured to compute attention weight. Furthermore, a local window is used instead of the global window in the attention mechanism to improve the local feature representation capabilities and reduce the computation. The proposed MF-DCMANet achieves competitive performance on the GOTCHA dataset, with a recognition accuracy of 99.75%. Furthermore, it maintains a high accuracy rate in the few-shot recognition and open-set recognition scenarios, outperforming the current state-of-the-art method by about 2%.

**Keywords:** PolSAR target; deep learning; feature fusion; transformer; Grassmann manifold

## 1. Introduction

It is well established that PolSAR target recognition has become increasingly significant in battlefield surveillance, air and missile defense, and strategic early warning, providing an important guarantee for battlefield situation awareness and intelligence generation [1]. Multi-polarization SAR data offers an advantage over single-polarization SAR data in that it not only provides amplitude (intensity) information but also records backward scattering information of the target under different polarization states, which can be represented through the polarimetric scattering matrix [2]. The polarimetric scattering matrix unifies the energy, phase, and polarization characteristics of target scattering, which are highly dependent on the target's shape, size, structure, and other factors [3]. It provides a relatively complete description of the electromagnetic scattering properties of the target. Therefore, it is essential to make reasonable use of fundamental or further processed polarization information to enhance the target recognition capability. However, in most studies, polarization information is often applied to terrain classification tasks that assign class labels to individual pixels in the image possessing semantic information. Zhou et al. [4] extracted a six-dimensional real-valued feature vector from the polarization covariance matrix and then fed the six-channel real images into a deep network to learn hierarchical polarimetric spatial features, achieving satisfactory results in classifying 15 terrains in the Flevoland data. Zhang et al. [5] employed polarization decomposition to crops in PolSAR scenes and

then fed the resulting polarization tensors into a tensor decomposition network for dimension reduction, which achieved better classification accuracy. However, these pixel-scale terrain classification methods cannot be directly applied to image-scale target recognition tasks. Therefore, for the PolSAR target recognition tasks, methods that exploit polarization information at the image scale should be developed.

Despite the promising results of terrain classification based on polarization features, the efficacy of utilizing a single feature to identify targets in a complex and dynamic battlefield environment is limited [6,7]. A single feature only portrays the target characteristics from one aspect, which makes it difficult to describe all the information embedded in the polarization target. The application of multi-feature fusion recognition methods allows for the comprehensive exploitation and utilization of diverse information contained in multi-polarization SAR data, effectively solving the problem of insufficient robustness of a single feature in complex scenarios [8–10]. Based on human perception and experience accumulation, researchers have designed many distinctive features from the intensity map of PolSAR targets, which generally have specific physical meanings. At present, various features have been developed for target recognition tasks, such as monogenic signals [11], computer vision features [12], and electromagnetic scattering features [13]. The potential feature extraction process of the monogenic signal has the characteristics of rotation-invariance and scale-invariance and has been widely investigated and explored in the domain of PolSAR target recognition. Dong et al. [14–16] and Li et al. [10] introduced monogenic signal analysis into the task of SAR target recognition, systematically analyzing the advantages of monogenic signal in describing SAR target characteristics. They also designed multiple feasible classification strategies to improve the target recognition performance. These handcrafted features have a strong discriminative ability and are not restricted by the amount of data, so they are more suitable for the PolSAR target recognition field with few labeled samples; however, they have difficulty in excavating deeper features of the image and lack universality. Moreover, the distinctive imaging mechanism of PolSAR, coupled with the diversity of target categories and the challenge of adapting to different datasets, makes it difficult to fully maximize the discriminative properties of SAR data. Therefore, artificial feature design remains a challenging task.

Lately, deep learning has greatly promoted the development of the computer vision field [17,18]. By leveraging neural networks to automatically discover more abstract features from input data, deep learning reduces the incompleteness caused by handcrafted features, leading to more competitive performance compared to traditional methods. Chen et al. designed a network [19] specifically for SAR images called A-ConvNet. The average accuracy rate of 10 types of target classification on the MSTAR dataset can reach 99%. The CV-CNN proposed in [20] uses complex parameters and variables to extract features from PolSAR data and perform feature classification, effectively utilizing phase information. The convolution operation in CNNs facilitates the learning and extraction of visual features, but at the same time, CNN also introduces inductive bias during the process of feature learning, which limits the receptive fields of the features. This results in CNN being adept at extracting effective local information but struggling to capture and store long-range dependent information. The recently developed Vision Transformer (ViT) [21,22] effectively addresses this problem. ViT models the global dependencies between input and output by utilizing the self-attention mechanism, resulting in more interpretable models. As a result, ViT has found applications in the field of PolSAR recognition.

To maximize the utilization of intensity and polarization information obtained from PolSAR data and take advantage of the great benefits of deep learning in feature extraction, we propose a robust multi-feature dual-stage cross manifold attention network, namely, MF-DCMANet, which consists of the Multi-Feature (MF) extraction, the Cross-Feature Network (CFN) and Cross-Manifold Attention (CMA) modules. In the multi-feature extraction stage, we extracted monogenic features and polarization features from the PolSAR target. In the first stage, the CFN module is used for performing cross-fusion feature extraction, followed by the CMA module in the second stage, thus merging features of different attributes

to generate deeper high-level semantic features. We conducted a comprehensive set of comparison experiments on the GOTCHA dataset.

Until now, there has been a scarcity of research on utilizing ViT for extracting features from PolSAR images in target recognition. This article makes the following contributions:

- A multi-feature extraction method specifically for PolSAR images has been proposed. The multi-feature extracted by this method can describe the target stably and robustly and is not affected by the target pose, geometry, and radar parameters as much as possible;
- A dual-stage feature cross-fusion representation framework is proposed, respectively named Cross-Feature Network (CFN) and Cross-Manifold Attention (CMA);
- In MF-DCMANet, handcrafted monogenic features and polarization features are combined with deep features to improve target recognition accuracy;
- By leveraging fusion techniques, the proposed MF-DCMANet enhances recognition performance and achieves the highest accuracy on the fully polarimetric GOTCHA dataset;
- It is often challenging to obtain sufficient and comprehensive samples in practical PolSAR target recognition applications. Despite this limitation, the proposed method still achieves satisfactory performance in few-shot and open-set recognition scenarios.

The organizational structure of this paper is as follows. Section 2 presents related works using CNN and Transformer models. Section 3 presents a detailed exposition of the proposed MF-DCMANet. Section 4 presents comparative experiments and discussion analysis, and Section 5 summarizes the content and innovations of the paper.

## 2. Related Works

### 2.1. CNN-Based Multi-Feature Target Recognition

The methods of multi-feature target recognition based on CNN can mainly be divided into two categories: one is the combination of deep features and handcrafted features, while the other is the combination of deep features learned from different layers of the network for classification.

In the work of combining deep features and handcrafted features, Xing et al. [8] fused the scattering center features and CNN features through discriminant correlation analysis and achieved satisfactory results under the extended operating conditions of the MSTAR dataset. Zhang et al. concatenated Hog features with multi-scale deep features for preferable SAR ship classification [23]. Zhou et al. [24] automatically extracted semantic features from the attributed scattering centers and SAR images through the network and then simply concatenated the features for target recognition. Note that in the above fusion methods, different features are extracted independently, and the classification information contained in the features is only converged in the fusion stage. Zhang et al. [25,26] utilized polarimetric features as expert knowledge for the SAR ship classification task, performed effective feature fusion through deep neural networks and achieved advanced classification performance on the OpenSARShip dataset. Furthermore, Zhang et al. [27] analyzed the impact of integrating handcrafted features at different layers of the deep neural networks on recognition rates and introduced various effective feature concatenation techniques.

To effectively use the features learned by different layers of the network, Guo et al. [28] used convolution kernels of different scales to extract features of different levels in SAR images. Ai et al. [29] used different sizes of convolutional kernels to extract from images and then combined them through weighted fusion. The weights were learned via the neural network and achieved good recognition results on the MSTAR dataset. Zeng et al. [30] introduced a multi-stream structure combined with an attention mechanism to obtain rich features of targets and achieved better recognition performance on the MSTAR dataset. Zhai et al. [31] introduced an attention module into the CNN architecture to connect the features extracted from different layers and introduced transfer learning to reduce the demand for the number of training samples.

The methods for multi-feature fusion described above primarily use concatenation to combine features, which may not be effective in merging features with different attributes

and can lead to weak fusion generalization. In this paper, we propose a novel multi-feature cross-fusion framework to enhance the capability of PolSAR target recognition.

### 2.2. Transformer in Target Recognition

CNN has a relatively large advantage in extracting the underlying features and visual structure. However, the receptive field of CNN is usually small, which is not conducive to capturing global features [21]. In contrast, the multi-head attention mechanism of the transformer is more natural and effective in handling the dependencies between long-range features. Dosovitskiy et al. [22] successfully applied the transformer to the visual field (ViT). ViT treats the input image as a series of patches, where channels are connected across all the pixels in the patch and then linearly projected to the desired input dimension, flattening each patch into a single vector. Zhao et al. [32] applied the transformer to the few-shot recognition problem in the field of SAR recognition, constructed a support set and query set from original MSTAR data, and then calculated the attention weight between them. The attention weight is obtained by computing cosine similarity in Euclidean space. Wang et al. [33] developed a method combining CNN and transformer, which makes full use of the local perception capability of CNN and the global modeling capability of the transformer. Li et al. [34] constructed a multi-aspect SAR sequence dataset from the MSTAR data. The convolutional autoencoder is used as the basic feature extractor, and the dependence between sequences is mined through the transformer. The method has good noise robustness and achieves higher recognition accuracy.

These transformer-based methods calculate the attention weight in Euclidean space, and the nonlinear relationship between data is not effectively utilized. In contrast to the traditional transformer that is based on Euclidean space, we extend the transformer to a new research field, i.e., PolSAR target recognition, and propose a novel transformer based on manifold space to mine the high-dimensional relationship among features.

### 3. Methods

The MF-DCMANet is a multi-feature cross-fusion framework, and Figure 1 provides an overview of the entire framework. The framework encompasses three main modules: multi-feature extraction, Cross-Feature Network (CFA) module, and Cross-Manifold Attention (CMA) module. This section provides a detailed elaboration of the MF-DCMANet.

### 3.1. Problem Formulation

Let $m \in \mathbb{R}^{H \times W \times C_1}$ and $\mathcal{P} \in \mathbb{R}^{H \times W \times C_2}$ denote the multi-scale monogenic features and polarization features extracted from PolSAR targets with $C_1$ and $C_2$ channels, respectively. The objective of MF-DCMANet is to conduct dual-stage cross-fusion on the multi-features extracted from the target, and fused features are fed into the SoftMax layer after dimension reduction to obtain the predicted label of the target:

$$y = \mathcal{F}\Big(\mathrm{CMA}\Big(\mathrm{CFN}\Big(\mathcal{F}_c^1(m), \mathcal{F}_c^2(\mathcal{P})\Big)\Big)\Big), \tag{1}$$

where $\mathcal{F}_c$ represents CNN-based feature extractor. CFN is the first-stage cross-fusion module, CMA is the second-stage cross-fusion module, $\mathcal{F}$ is the prediction network, and $y$ is the predicted label.

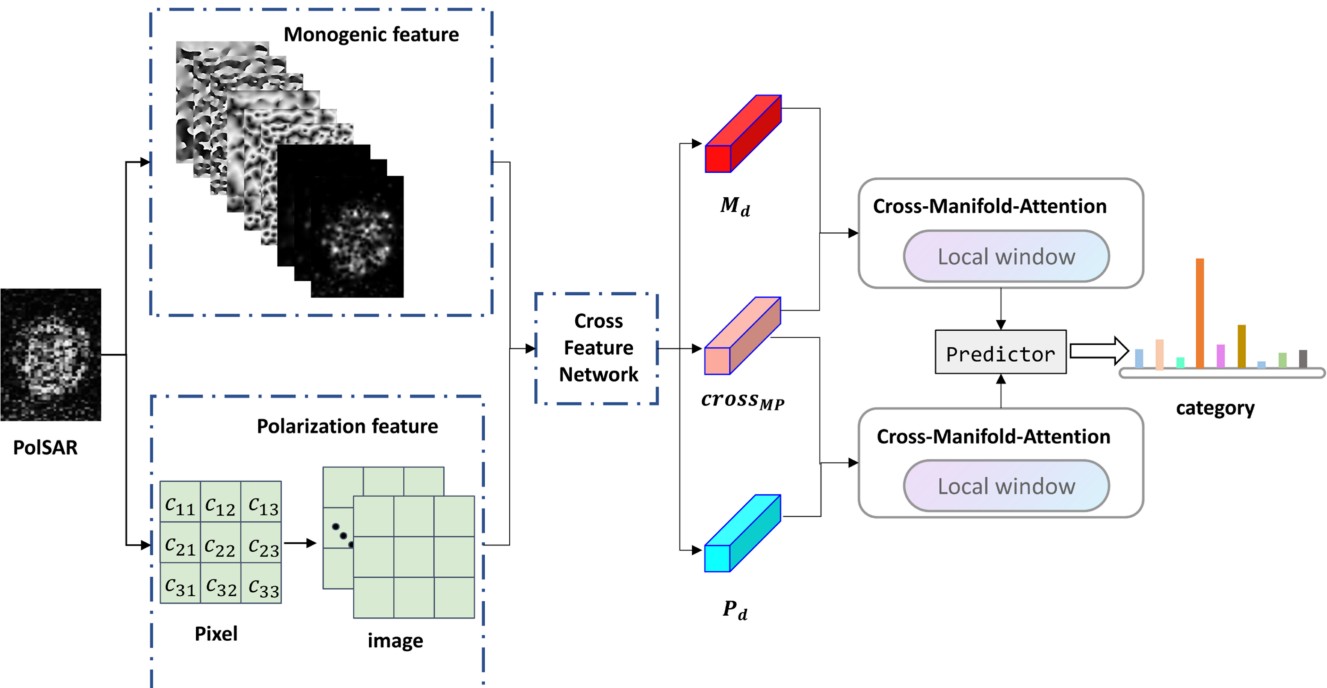

**Figure 1.** The overview of the entire framework. First, we extract low-level features from the intensity image and multi-polarization channels of POLSAR data, which are monogenic features $m$ and polarization features $\mathcal{P}$, respectively, and then use two fully convolutional neural networks to mine the mid-level semantic features contained in the polarization features and monogenic features. The extracted mid-level semantic features are fed to the first-stage cross-feature network (CFN) to obtain the fused features, followed by feeding the fused features ($cross_{MP}$) and mid-level semantic features ($M_d, P_d$) into the second-stage cross-manifold-attention (CMA) transformer. In the CMA module, these features are first encoded as tokens and then represented on the Grassmann manifold to mine the nonlinear correlation between features, which are mutually supplemented through multiple attention fusions.

### 3.2. Multi-Feature Extraction

#### 3.2.1. Monogenic Feature Extraction

The monogenic signal is a high-dimensional analytic signal derived from the one-dimensional analytic signal, which extracts rotation- and scale-invariant features from intensity images. The Hilbert transform can construct an analysis signal representation of a one-dimensional (1D) signal. Similarly, the Riesz transform can construct an analysis signal representation of a multi-dimensional signal. This article focuses on discussing the two-dimensional (2D) finite-length signal (image); that is, the dimension in the Riesz transform is 2 [35]. The Riesz kernel space $\left(h_x, h_y\right)$ of the original signal in the $x$-direction and $y$-direction is expressed as:

$$\left(h_x, h_y\right) = \left(\frac{x}{2\pi ||x||^3}, \frac{y}{2\pi ||y||^3}\right), \tag{2}$$

In practical applications, the image signal is a finite-length two-dimensional signal [36], and therefore, it is essential to employ a band-pass filter to extend the space domain infinitely. To maintain the odd symmetry of the Riesz transform kernel [37], the band-pass filter must satisfy the symmetry condition. Here the Log-Gabor filter is applied to achieve the effect of band-pass filtering. The 2D monogenic signal $I_M$ corresponding to image $I_0 \, \epsilon \mathbb{R}^{H \times W}$ can be mathematically represented as follows:

$$I_M = \left(I, I_x, I_y\right) = \left(I, h_x * I, h_y * I\right), \; I = I_0 * F^{-1}(G(\omega)), \tag{3}$$

where $F^{-1}$ represents the inverse transform of the two-dimensional Fourier transform, $I$ is an extension of $I_0$, $I_x$ and $I_y$ respectively represent the Riesz transform of $I$ in the $x$ and $y$ direction. The operator $*$ denotes convolution. $G(!)$ represents the frequency response of the Log-Gabor filter as follows:

$$G(\omega) = exp\left\{ -[log(\omega/w_0)]^2 / (2*[log(\sigma/w_0)]^2) \right\}, \ \omega_0 = \left( \lambda_{min}\mu^{S-1} \right)^{-1}, \quad (4)$$

where $S$ ($S = 1, 2, 3$) is the scale level index, with $S$ being an integer; $\omega_0$ is the central frequency; $\sigma$ is the broadband proportional factor; $\lambda_{min}$ is the minimum wavelength; $\mu$ expresses the wavelength multiplication coefficient. Next, the local amplitude $A$, local orientation $\theta$, and local phase $P$ of the input image can be defined as:

$$A = \sqrt{I^2 + I_x^2 + I_y^2}, \quad (5)$$

$$\theta = arctan\frac{I_y}{I_x}, \ \theta \in \left( -\frac{\pi}{2}, \frac{\pi}{2} \right], \quad (6)$$

$$P = \arctan\left( \left| \sqrt{I_x^2 + I_y^2} \right|, I \right), \ P \in (-\pi, \pi], \quad (7)$$

The $\mathcal{L}$-scale monogenic features $\{ I_M^1, I_M^2, \ldots, I_M^{\mathcal{L}} \}$ are obtained by changing the scale level index $S$ as:

$$m = \underbrace{\{ A_1, \theta_1, P_1}_{I_M^1}, \underbrace{A_2, \theta_2, P_2}_{I_M^2}, \cdots, \underbrace{A_{\mathcal{L}}, \theta_{\mathcal{L}}, P_{\mathcal{L}} \}}_{I_M^{\mathcal{L}}} \in \mathbb{R}^{H \times W \times 3\mathcal{L}}, \quad (8)$$

where $\mathcal{L} = 3$. Figure 2 shows the monogenic feature maps of the PolSAR target when $S$ is 1, 2, and 3, respectively.

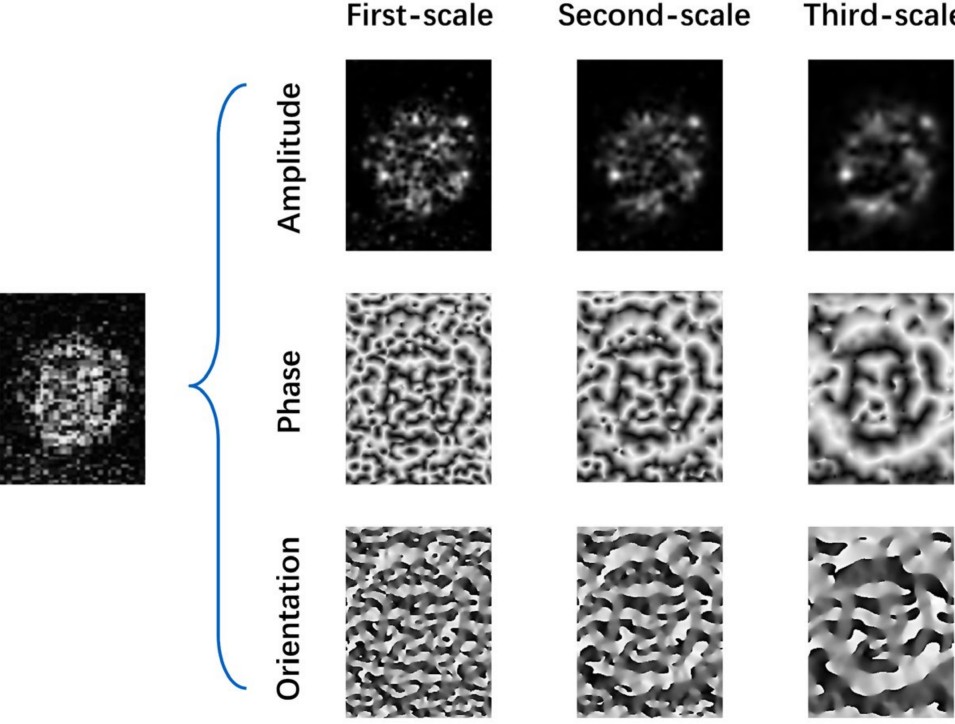

**Figure 2.** The three-scale monogenic features of a PolSAR target.

### 3.2.2. Polarization Feature Extraction

In the single base station PolSAR measurement, considering the reciprocity theorem, the cross-polarization component $S_{hv} = S_{vh}$ and Equation (9) defines the polarization scattering vector of the target, where $h$ and $v$ respectively denote the electromagnetic waves of radar transmission in the horizontal and vertical polarizations, $T$ represents the transpose operation [38].

$$k = \left[ S_{hh}, \sqrt{2} S_{hv}, S_{vv} \right]^T, \tag{9}$$

For PolSAR data after multi-view processing, each pixel can be expressed by a covariance matrix $C$ with the following expression:

$$C = \frac{1}{N} \sum_{i=1}^{N} k_i k_i^H \in \mathbb{C}^{3 \times 3}, \tag{10}$$

where $N$ is the image view number and $H$ represents the Hermitian transpose operation.

The polarization covariance matrix is a symmetric matrix, with all of its elements being complex numbers except for the diagonal elements. To highlight the capability of the cross-fusion module in feature extraction and fusion, we simply transform the covariance matrix into a nine-dimensional real vector $[C_{11}, C_{22}, C_{33}, C_{12r}, C_{12i}, C_{13r}, C_{13i}, C_{23r}, C_{23i}]$ based on maximizing the preservation of covariance elements. The polarimetric features extracted from the input image $I_0 \in \mathbb{R}^{H \times W}$ are expressed as polarimetric tensors $\mathcal{P} \in \mathbb{R}^{H \times W \times 9}$.

The distribution of sample-averaged elements intensities of the nine-dimensional polarization feature vector is presented in Figure 3 with box plots. Note that the $y$ values are given in a relative form ranging from 0 to 1. We noticed that the intensity distribution of polarization features of different target samples lies in different intervals, exhibiting a certain degree of separability. For example, the $C_{12r}$ mean values of different target samples have a small span and do not overlap significantly. Therefore, instead of learning features directly from the original PolSAR targets, we further utilize CNN to automatically learn more discriminative mid-level semantic features from the basic polarization features to improve recognition performance.

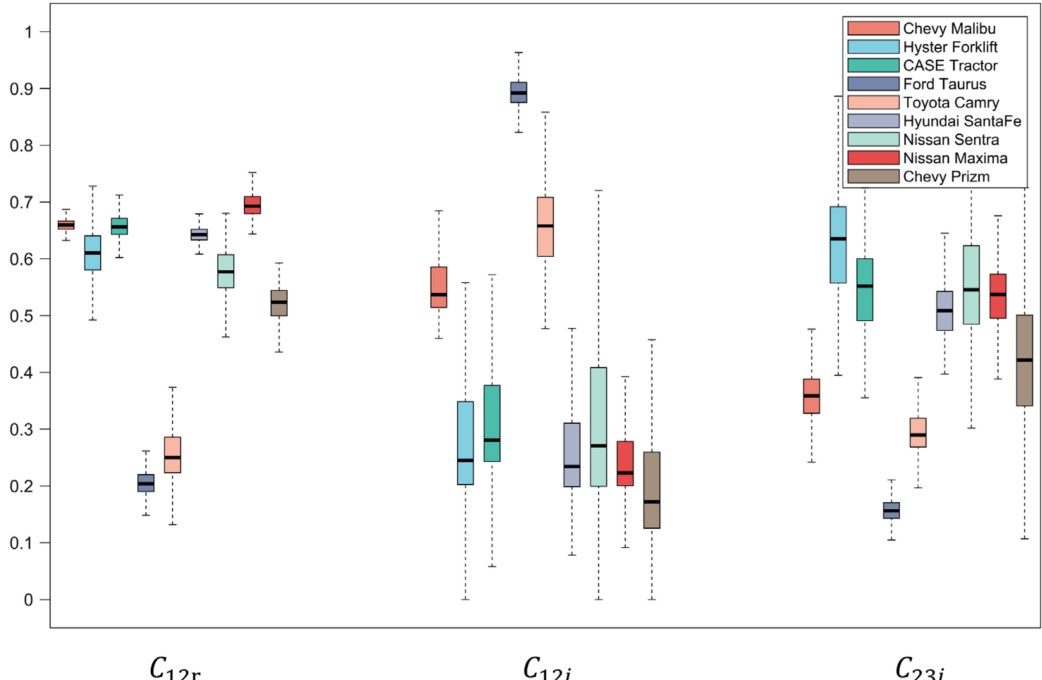

**Figure 3.** Distribution of relative polarization feature intensities in different target categories.

In the multi-feature extraction stage, we extracted multi-scale monogenic features from the intensity image of the PolSAR target and polarization features from the multi-polarization channels. The monogenic signal characterizes the scattering phenomenon of PolSAR targets and has the characteristics of rotation-invariance and scale-invariance. The polarization features extracted from the polarization covariance matrix contain all the target polarization information obtained from the radar measurement, which can finely describe the target scattering mechanism from the physical meaning.

### 3.3. Cross-Feature-Network (CFN)

The performance of convolutional neural networks has been widely verified on various deep learning tasks [39–45]. Due to the advantages of CNN in feature extraction, we use CNN to mine middle-level semantic information for low-level features such as monogenic feature tensor and polarization feature tensor extracted in the multi-feature extraction stage. Compared with directly extracting features from PolSAR targets, these well-designed handcrafted features contain richer information, which can be directly input into the network for secondary information extraction.

This section designs a cross-feature fusion network named CFN. The CFN module consists of two Ex-nets and a Fu-net, Table 1 lists the configurations of the network, where Conv, BN, and MP are Convolution, Batch normalization, and Max Pooling, respectively, while *d* is the number of channels of the input feature. Also, the output size of each block is indicated by the last component in that block. The process of fusion in the CFN module is depicted in Figure 4. With the forward propagation process of the network, the weights of different feature extraction networks (Ex-nets) are exchanged to mine the correlation information between features, and the classification information carried by different features are effectively and compactly fused.

**Table 1.** Configurations of the CFN module.

| Input | Ex-Net | | | | Fu-Net |
|---|---|---|---|---|---|
| | **Block1** | **Block2** | **Block3** | **Block4** | **Block5** |
| | $5 \times 5$ Conv | $5 \times 5$ Conv | $5 \times 5$ Conv | $5 \times 5$ Conv | $1 \times 1$ Conv |
| | BN | BN | BN | BN | BN |
| $20 \times 20 \times d$ | Relu | Relu | Relu | Relu | Relu |
| | | $2 \times 2$ MP | | | |
| | $20 \times 20 \times 16$ | $10 \times 10 \times 32$ | $10 \times 10 \times 64$ | $10 \times 10 \times 128$ | $10 \times 10 \times 128$ |

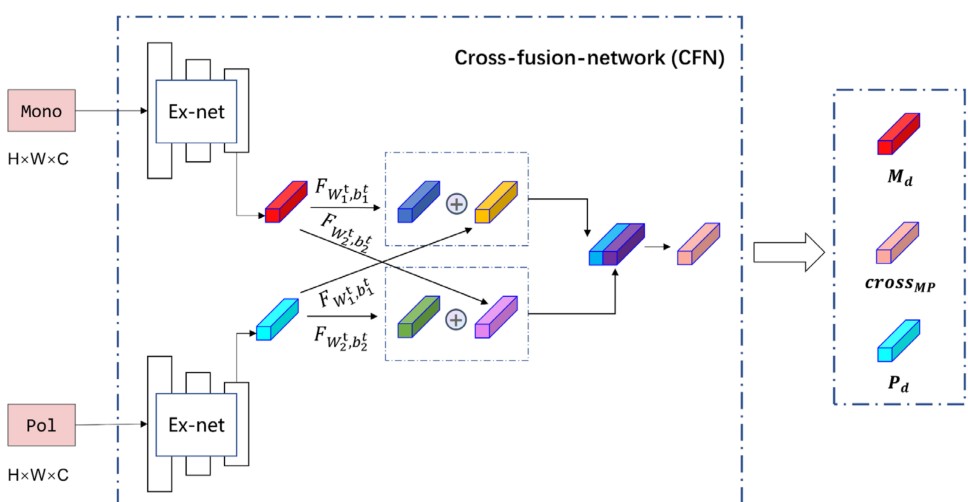

**Figure 4.** The CFN framework.

Once the monogenic feature $m$ and polarization feature $\mathscr{P}$ pass through the Ex-net, the output in the $t$-th layer of the Ex-net can be formulated as $\left\{ m_1^t, \cdots m_N^t \right\}$, $\left\{ \mathscr{P}_1^t, \cdots \mathscr{P}_N^t \right\}$, where $N$ represents the training sample size, and $t$ represents the last layer of the Ex-net. We can consider the output of the Ex-net as the updated input, which can then be input into the Fu-net. For instance, in the case of the $i$-th image, the fusion representation can be described as follows:

$$m_i^{t+1} = F_{W_1^t, b_1^t}\left( m_i^t \right) + F_{W_1^t, b_1^t}\left( \mathscr{P}_i^t \right), \tag{11}$$

$$\mathscr{P}_i^{t+1} = F_{W_2^t, b_2^t}\left( \mathscr{P}_i^t \right) + F_{W_2^t, b_2^t}\left( m_i^t \right), \tag{12}$$

$$v_i^{t+1} = \left[ m_i^{t+1}, \mathscr{P}_i^{t+1} \right], \tag{13}$$

where $t + 1$ represents the first layer of Fu-net, $F(\cdot)$ is the nonlinear mapping function concerning the learnable weights $W$ and biases $b$ of all layers in the Ex-net and Fu-net, The subscripts 1 and 2 denote the monogenic and polarization feature extraction networks, respectively.

The outputs of the CFN module are mid-level monogenic features, mid-level polarization features, and cross-fusion features, which are named $M_d$, $P_d$, and $cross_{MP}$, respectively.

### 3.4. Cross-Manifold-Attention (CMA)

The latest advances in machine learning and computer vision [46] show that non-Euclidean-based subspace learning (e.g., Grassmann manifold space) is more conducive to solving multi-classification problems in various fields. Compared with the conventional self-attention mechanism, CMA transforms the feature vectors of Euclidean space onto the Grassmann manifold to form feature patches and measures the similarity between feature patches through the distance measure defined on the Grassmann manifold. Figure 5 presents the framework of CMA.

As shown in Figure 5, to ensure that $M_d$ and $P_d$ are not limited to the information carried by themselves, the CMA module takes $cross_{MP}$ as the rich prior in the form of keys/values and take $M_d$ or $P_d$ as query, then the CMA module represents key and query on the Grassmann manifold and performs multi-head attention to refine input feature patches via cross-attention. To reduce the computational cost and hardware requirements, we use local windows [47] instead of global windows in the self-attention mechanism to enhance the ability to represent local features. The global window allows each pixel to attend to every other pixel, whereas the local window localizes attention for each pixel to a neighborhood around itself. Therefore, each pixel's attention span is usually different from the next. Specifically, given input feature tensors $M_d, cross_{MP} \in \mathbb{R}^{H_0 \times W_0 \times C_0}$ with height $H_0$, width $W_0$, and $C_0$ channels. The input feature is partitioned into non-overlapping rectangular patches of size $S \times S$ in ViT and these patches are projected linearly into the $D$-dimensional hidden space. The result is a set of vectors of patch embeddings $M_d, cross_{MP} \sim X \in \mathbb{R}^{L \times D}$, where $L = H_0 W_0 / S^2$ is the sequence length.

Given query $Q \in \mathbb{R}^{L \times D}$ and key $K \in \mathbb{R}^{L \times D}$, under the self-attention mechanism, the attention weight $A$ is computed based on the pairwise cosine similarity between the query $Q_i \in \mathbb{R}^{1 \times D}$ and key $K_j \in \mathbb{R}^{1 \times D}$ on the Euclidean manifold,

$$A(Q, K) = softmax\left( \frac{1}{\sqrt{D}} Q K^T \right) \in \mathbb{R}^{L \times L}. \tag{14}$$

The cosine similarity evaluates the similarity between two vectors by computing the cosine value of the angle between them, which varies between +1 (most similar) and −1 (least similar).

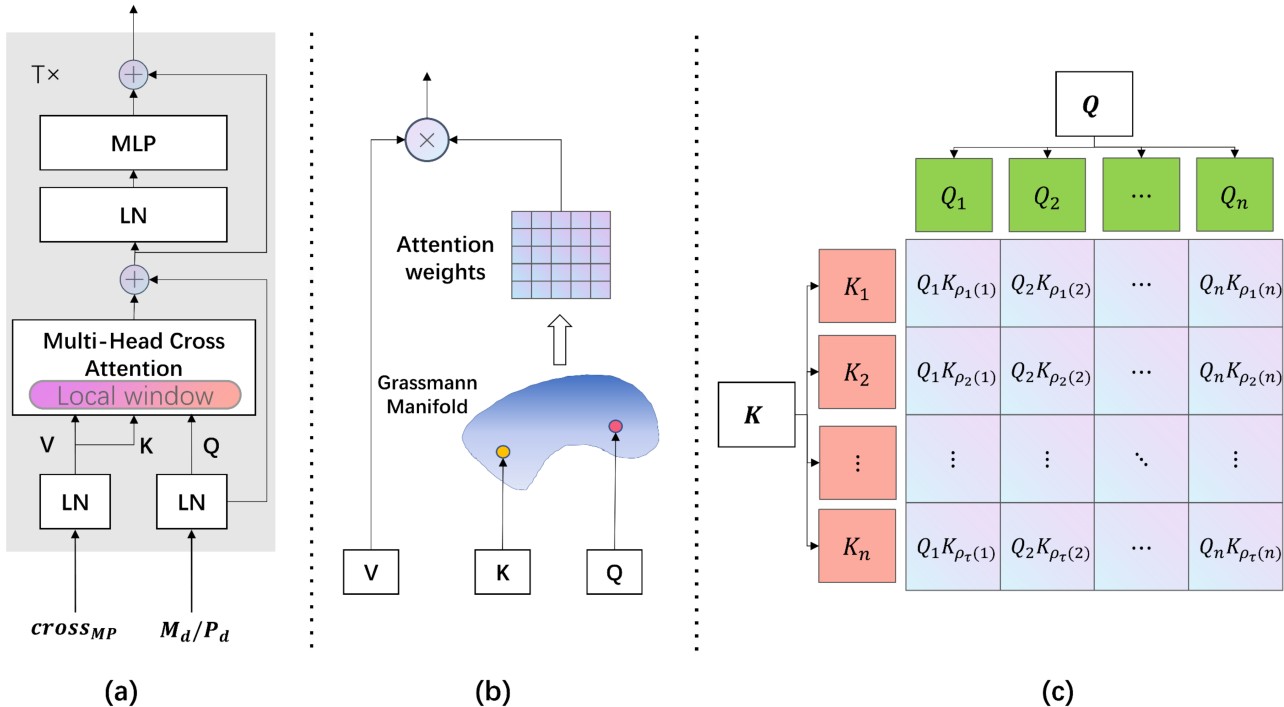

(a)　　　　　　　　　　　(b)　　　　　　　　　　　(c)

**Figure 5.** The CMA framework. (**a**) Transformer encoder with local window. (**b**) Cross-manifold-attention mechanism. (**c**) The calculation process for attention weight, where $Q_1 K_{\rho_1(1)}$ denotes the projection distance between $Q_1$ and $K_{\rho_1(1)}$ calculated according to Equation (17), $\rho_\tau(i)$ denotes $i$'s $\tau$-th neighbor.

Next, we convert the attention weight calculation process from the Euclidean space to the Grassmann manifold space in two steps: (1) represent the features onto the Grassmann manifold (see Section 3.4.1); (2) calculate the attention weight using the projection distance (see Section 3.4.2).

### 3.4.1. Feature Representation on Grassmann Manifold

Equation (15) shows the mathematical definition of the Grassmann manifold [48], which is composed of an orthonormal basis of $n \times k$ dimensional matrix X.

$$\mathcal{G}(n,k) = \left\{ span(X) : X \in \mathbb{R}^{n \times k}, XX^T = I_k \right\}, \tag{15}$$

where $span(X) \in \mathcal{G}(n,k)$ denotes the subspace spanned by columns of X. Given query $Q \in \mathbb{R}^{L \times D}$ and key $K \in \mathbb{R}^{L \times D}$, we first expand the $D$-dimensional features to a two-dimensional plane to form a feature patch, although the column vectors of $Q_i \in \mathbb{R}^{n \times k}$, $i = 1, \cdots, L$ and $K_j \in \mathbb{R}^{n \times k}, j = 1, \cdots, L$ are not orthogonal, it is easy to implement the orthogonalization constraint with the help of mathematical techniques such as Gram-Schmidt orthogonalization [49,50] so that the $Q_i$ and $K_j$ are considered as elements of the Grassmann manifold $G(n,k)$, where $D = n \times k$.

### 3.4.2. Distance Metrics on Grassmann Manifold

The normalization metric between two elements $Q_i, K_j \in \mathcal{G}(n,k)$ can be obtained by computing the principal angles $\{\theta_l\}_{l=1}^k$ between them [41]. The principal angle can be described mathematically by

$$\begin{cases} cos\theta_l = \max\limits_{u_l \in span(Q_i)} \max\limits_{v_l \in span(K_j)} u_l^T v_l, \\ \text{subject to } u_l^T u_l = 1, v_l^T v_l = 1, for\ l = 1, \cdots, k \\ \quad \text{subject to } u_l^T u_m = 0, v_l^T v_m = 0, \forall\ m < l. \end{cases} \tag{16}$$

In practice, the principal angles between $Q_i, K_j \in \mathcal{G}(n, k)$ are calculated by singular value decomposition, where the singular values of $Q_i^T K_j$ are the cosines of the principal angles. Based on the definition of the principal angle, scholars have proposed various Grassmann manifold distances, such as Projection distance, Binet-Cauchy distance, and Procrustes distance [51,52]. The projection distance is the sum of the square of the sine of the principal angle:

$$d_{pj}^2 (Q_i, K_j) = \sum_{l=1}^{k} sin^2(\theta_l) = m - \sum_{l=1}^{k} cos^2(\theta_l) = \left\| Q_i Q_i^T - KK_j^T \right\|_F^2, \tag{17}$$

where $||\cdot||_F$ denotes the matrix Frobenius norm. The CMA module uses a $w \times w$ local window instead of a $\sqrt{L} \times \sqrt{L}$ global window, we define attention weight for the $i$-th input $Q_i$ with neighborhood window size $w^2$, $A_i^\tau$ as the projection distance of $Q_i$ and $K_j$:

$$A_i^\tau = softmax \left( \frac{1}{\sqrt{k}} \begin{bmatrix} ||Q_i Q_i^T - K_{\rho_1(i)} K_{\rho_1(i)}^T ||_F \\ ||Q_i Q_i^T - K_{\rho_2(i)} K_{\rho_2(i)}^T ||_F \\ \vdots \\ ||Q_i Q_i^T - K_{\rho_\tau(i)} K_{\rho_\tau(i)}^T ||_F \end{bmatrix} \right), \; i = 1 : L, \; \tau = 1 : w^2 \tag{18}$$

where $\sqrt{k}$ is the scaling factor, $\rho_\tau(i)$ denotes $i's$ $\tau$-th neighbor, The dimension of attention weight $A$ is $\mathbb{R}^{L \times w^2}$, correspondingly, the dimension of value V corresponding to $Q_i$ is $\mathbb{R}^{w^2 \times D}$. By converting the feature vectors into feature patches and representing them on the Grassmann manifold, we fully exploit the nonlinear geometric characteristics inside the data.

The above procedures are formulated as

$$x_0 = \left[ x_p^1 E; x_p^2 E; \cdots ; x_p^N E \right], \; E \in \mathbb{R}^{(S^2 * c_0) \times D} \tag{19}$$

$$z_0 = \left[ z_p^1 E; z_p^2 E; \cdots ; z_p^N E \right], \tag{20}$$

$$x'_\updownarrow = CMA \left( LN \left( x_{\updownarrow - 1} \right), LN(z_0) \right) + x_{\updownarrow - 1}, \; \updownarrow = 1, \cdots, T \tag{21}$$

$$x_\updownarrow = MLP \left( LN \left( x'_\updownarrow \right) \right) + x'_\updownarrow, \tag{22}$$

$$y = LN(x_\updownarrow), \tag{23}$$

where $E$ represents the linear projection layer for mapping patches to a $D$-dimensional latent embedding space, $x_p^i, z_p^i \in \mathbb{R}^{1 \times (S^2 \cdot c_0)}$. Due to its special design, the CMA module can capture the local structure and long-range relationship of features, and the specific calculation follows the formulas below:

$$Q^\mu = \text{orth}\left(LN(z_0) E_q\right), \tag{24}$$

$$K^\mu = \text{LW}(\text{orth}(LN(x_\iota) E_k)), \tag{25}$$

$$V^\mu = LW(LN(x_\iota) E_v), \tag{26}$$

$$Att^\mu(Q^\mu, K^\mu, V^\mu) = AV, \mu = 1, 2, \cdots, H \tag{27}$$

$$x'_\iota = concat \left( \{ Att^\mu(Q^\mu, K^\mu, V^\mu) \}_{\mu=1}^{H} \right) E_{out} \tag{28}$$

where $E_q, E_k, E_v \in \mathbb{R}^{D \times D_h}$, $orth(\cdot)$ denotes the Gram-Schmidt orthogonalization, $LW(\cdot)$ indicates a local windowing operation, $E_{out} \in \mathbb{R}^{(H \cdot D_h) \times D}$, $H$ is the head number of multi-head attention. To keep compute and number of parameters constant when changing $H$, $D_h$ is typically set to $D/H$.

### 3.5. Predictor

The prediction network concats the output features of the two CMA modules in the channel dimension. The primary purpose of the prediction network is to reduce the dimensionality of the features outputted by the CMA module into a nine-dimensional vector that is used for predicting the target labels. Figure 6 details the structure of the prediction network.

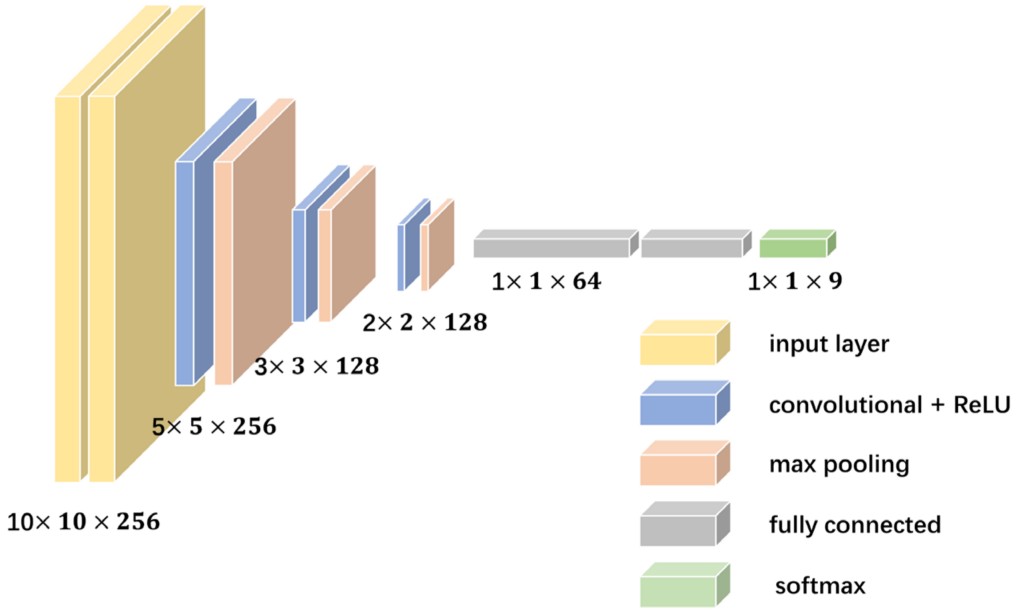

**Figure 6.** The Prediction network architecture.

### 4. Experiments

In this chapter, we conducted numerous target recognition experiments using the GOTCHA measurement dataset to perform validation of the rationality of the proposed method and verified the feasibility and scalability of the proposed MF-DCMANet through comparison with other advanced methods.

The experiment hardware configuration includes an AMD Ryzen 5800H CPU, Nvidia RTX3060 GPU, and 16GB memory. The software environment is composed of Matlab 2021b, Python 3.7, CUDA 11.2, and Tensorflow 2.11. The monogenic and polarization features are extracted using Matlab 2021b, and the network training is completed in the Python environment.

### 4.1. Data Description

To support the study, a GOTCHA dataset for PolSAR target recognition is constructed using raw data from the "Gotcha Volumetric SAR Data Set V1.0" [53]. The GOTCHA dataset is an airborne 360° circular SAR dataset constructed by researchers from AFRL and Ohio State University for numerous civilian vehicles parked in the internal parking lot [54]. The original dataset comprises SAR phase history data that were obtained in the X-band, possessing a bandwidth of 640 MHz and consisting of data collected at eight different elevation angles (eight circular passes). The raw data of all flight passes are stored in Matlab binary form under eight folders. Each folder contains four corresponding polarization channel data (HH, HV, VH, and VV polarization channels, respectively),

and each polarization channel contains pulse data of the whole parking lot from 0 to 360 degrees azimuth.

We selected regions of interest containing the desired targets from the original scene, as shown in Figure 7. The optical images of the targets of interest are presented in Figure 8.

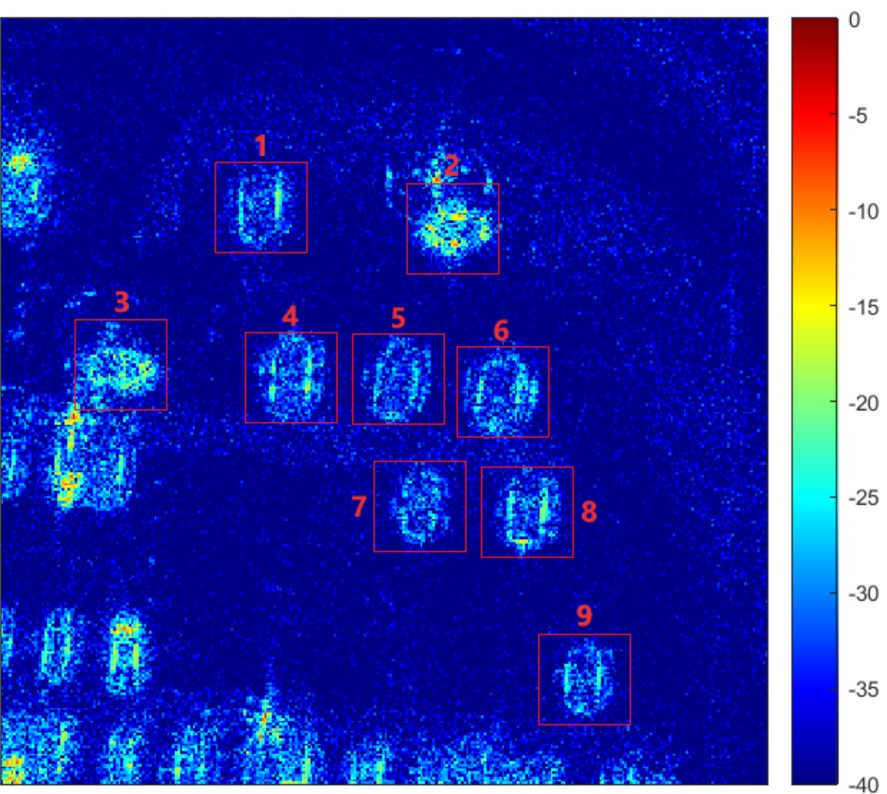

**Figure 7.** Scene regions containing nine categories of targets of interest.

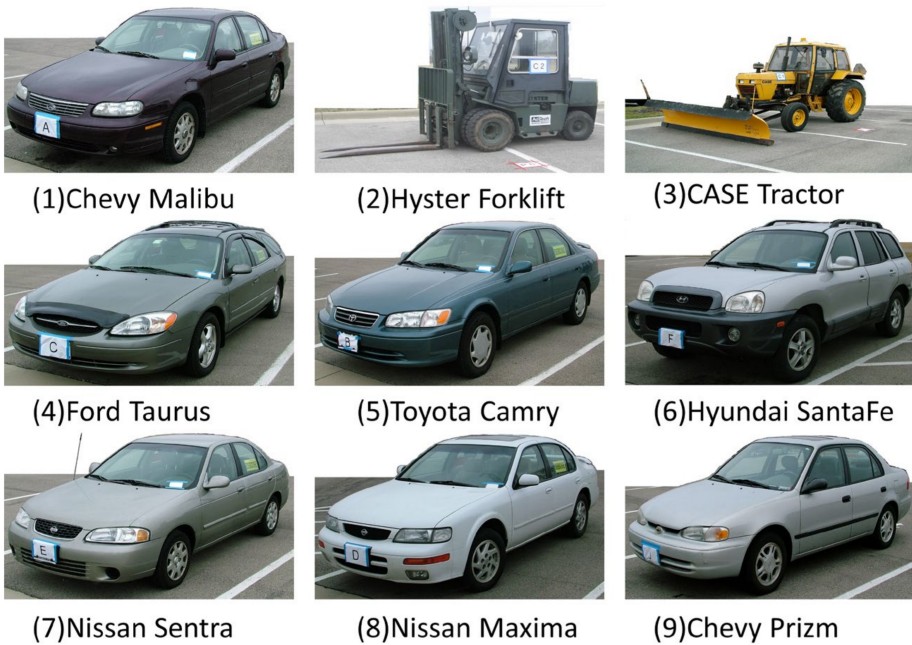

(1)Chevy Malibu (2)Hyster Forklift (3)CASE Tractor
(4)Ford Taurus (5)Toyota Camry (6)Hyundai SantaFe
(7)Nissan Sentra (8)Nissan Maxima (9)Chevy Prizm

**Figure 8.** Optical images of vehicle targets in the GOTCHA dataset.

To construct a dataset suitable for PolSAR target recognition from scene imaging maps, the $360°$ circular aperture was partitioned into 90 sub-apertures with $4°$ azimuth angles. The four polarization channels under each circular pass are then superimposed to generate 90 imaging maps containing regions of interest with targets. Then, based on the positions of the nine target categories in the scene, we extracted PolSAR images of these nine target categories from the scene imaging maps, uniformly cropped to a size of $50 \times 50$.

So far, each circular pass contains nine categories of targets, with 90 images per category. We select the targets from pass1, pass3, pass5, and pass7 for training and the targets from the remaining passes for testing. The composition of the dataset is presented in Table 2.

**Table 2.** The composition of the dataset.

| Dataset | Category | Pass | Number |
|---------|----------|------|--------|
| Training set | 1~9 | 1, 3, 5, 7 | $360 \times 9$ |
| Test set | 1~9 | 2, 4, 6, 8 | $360 \times 9$ |

*4.2. Implementation Details*

First of all, we perform a unified Z-Score normalization operation on the GOTCHA dataset, which can be defined as: $x^* = (x - \bar{x})/\sigma$, where $\bar{x}$ and $\sigma$ represent the mean and standard deviation of an image $x$, respectively. Its primary objective is to normalize the data to the same magnitude and solve the comparability problem between the data. After the data is normalized, the network can converge to the optimal solution more quickly. For GOTCHA data, each polarization channel is independently normalized.

Prior to the monogenic feature extraction, the intensity image is formed by taking the complex-valued images of the four polarization channels as amplitude values. To speed up network training, the image size is compressed to $20 \times 20$ in the experiment. We extract the three-scale monogenic feature tensor from the intensity image. In Formula (4), the scale level index $S$ of the monogenic signal is set to 1, 2, 3; in turn, the broadband proportional factor $\sigma$ is set to 0.48, the minimum wavelength $\lambda_{\min}$ is set to 8, and the wavelength multiplication coefficient $\mu$ is set to 2.5. Then, the monogenic features obtained from the intensity image are arranged as a feature tensor $m \in \mathbb{R}^{20 \times 20 \times 9}$.

We extract the polarization covariance matrix for every pixel in the image of the GOTCHA data and convert it to a nine-dimensional real vector to form the polarization feature tensor $\mathcal{P} \in \mathbb{R}^{20 \times 20 \times 9}$.

The training parameter settings in the MF-DCMANet are shown in Table 3. The outputs of the CFN module are: $M_d, P_d, cross_{MP} \in \mathbb{R}^{10 \times 10 \times 128}$. In the CMA module, the local window size $w$ is fixed as 3, and the head number $H$ of multi-head attention is configured as 4. In the CMA module, we set the parameters on the Grassmann manifold $\mathcal{G}(n, k)$ to $n = 8$, $k = 4$. The parameters $n$ and $k$ can be adjusted according to the actual input size and only need to satisfy the condition $n \geq k > 0$.

**Table 3.** Training settings on the MF-DCMANet.

| | |
|---|---|
| Batch size | 64 |
| Optimizer | Adam |
| Initialized learning rate | 0.01 |
| Learning Rate Decay | Exponential-decay |
| Momentum | 0.9 |
| Weight decay | 0.0001 |
| Epochs | 100 |

### 4.3. Evaluation Metrics

4.3.1. Overall Accuracy (OA)

Overall accuracy (OA) refers to the proportion of correctly classified samples to the total samples:

$$\mathrm{OA} = \sum_{i=1}^{C} N_{ii} \Big/ \sum_{i=1}^{C} \sum_{j=1}^{C} N_{ij}, \tag{29}$$

where $C$ represents the total of all categories, $N_{ij}$ represents the number of samples that belong to category $i$ but are misclassified to be category $j$ and $N_{ii}$ is the quantity of samples being correctly classified.

4.3.2. Receiver Operation Characteristics (ROC)

The ROC curve [55] is an important and common statistical analysis method in machine learning. It is a graphical representation of the relationship between the true positive rate (TPR) and the false positive rate (FPR) for a given classification model. If two ROC curves intersect, it is difficult to assert their relative performance in a general sense. Therefore, the AUC value is introduced as a measure of the overall performance of the model. AUC can be obtained by integrating the ROC curve, and the value of AUC is usually between 0.5 and 1. A larger AUC represents better performance. TPR and FPR can be calculated by the following formulas:

$$\mathrm{TPR} = \mathrm{TP}/(\mathrm{TP} + \mathrm{FN}), \tag{30}$$

$$\mathrm{FPR} = \mathrm{FP}/(\mathrm{FP} + \mathrm{TN}). \tag{31}$$

### 4.4. Quantitative Analysis

The proposed MF-DCMANet is evaluated on the fully polarimetric GOTCHA datasets. The ablation experiments are used to study the impact of crucial components of the proposed MF-DCMANet on the overall performance.

4.4.1. Classification Results and Analysis

Figure 9 displays the confusion matrix of MF-DCMANet on the training set divided according to different proportions, allowing for a straightforward assessment of the proposed method's ability to classify different categories. The diagonal values represent elements where the predicted value equals the true value, while the off-diagonal values represent elements where the classifier made an incorrect prediction. A higher ratio of values on the diagonal compared to the off-diagonal indicates better performance by the classifier. Hence, in Figure 9, the darker (red) the diagonal color, indicating the better the performance of the model.

The diagonal elements in Figure 9a are close to 360, indicating that almost all samples were correctly classified, with some targets achieving 100% accuracy. In Figure 9b–d, where the data used for training is divided into 1/3, 1/7, and 1/10 of the complete training dataset, most of the test samples are still accurately classified on the diagonal line, but some test samples are mistakenly classified into off-diagonal lines, indicating that some targets have been misclassified as other types. Specifically, in Figure 9b–d, the most commonly confused targets are the first and seventh categories, the fifth and fourth categories, and the fifth and ninth categories. Optical images corresponding to these targets in Figure 8 demonstrate that these easily confused targets have similar appearances, resulting in reduced generalization ability of the extracted features in scenarios with limited training samples, making it easier to misclassify these targets with similar appearances.

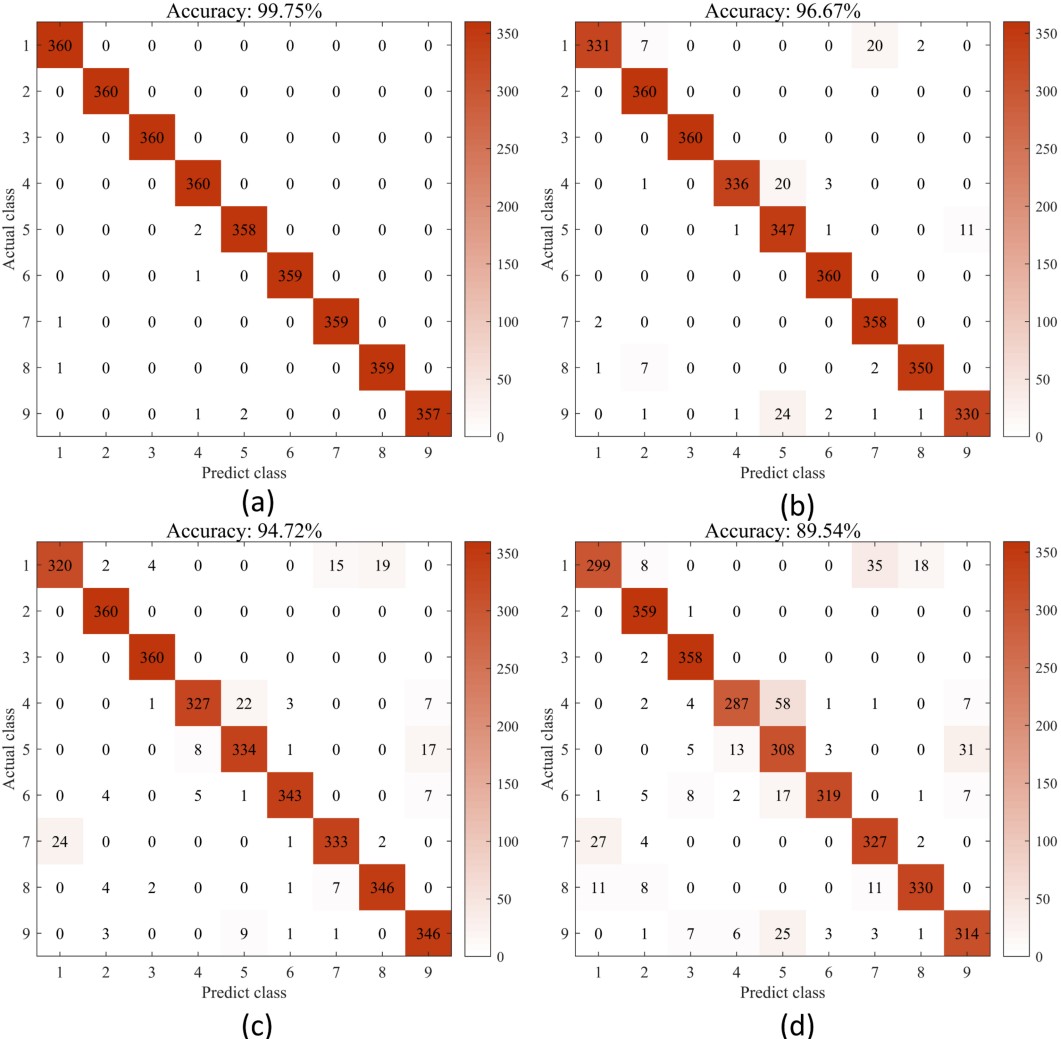

**Figure 9.** Confusion matrix of the MF-DCMANet on the GOTCHA dataset. (**a**) Full training datasets. (**b**) One-third of the training set. (**c**) One-seventh of the training set. (**d**) One-tenth of the training set.

To assess the effectiveness of the MF-DCMANet, a range of advanced PolSAR target recognition methods are compared as benchmarks, such as algorithms based on handcrafted features or deep learning and algorithms based on multi-feature fusion. Among the algorithms based on handcrafted features, we focus on comparing the methods based on monogenic features and polarization features, such as polarimetric scattering coding [56], polarimetric decomposition [57], Monogenic Scale Space (Mono) [14,58], Mono-HOG [59], Mono-BoVW [10], Monogenic Signal on Grassmann Manifolds (Mono-Grass) [15], and other methods, such as Steerable Wavelet Frames [16], Attributed scattering center (ASC) model [60]. Among deep learning-based algorithms, we compare the CNN-based and transformer-based methods, as well as the other novel methods. CNN-based methods include A-ConvNet [19], CV-CNN [20], CV-FCNN [61], CVNLNet [10], and RVNLNet with real input, the transformer-based method include ViT transformer [35], SpectralFormer [62], CrossViT [63], and other methods such as SymNet [64], monogenic ConvNet layer [65]. Furthermore, we also include multi-feature-based Mono-CVNLNet [10] and FEC [8] in our comparison of methods. The corresponding results and necessary descriptions of different methods are presented in Table 4.

**Table 4.** Performances of different methods in the GOTCHA dataset.

| Input | Method | | Classifier | OA (%) | FPS |
|---|---|---|---|---|---|
| Handcrafted features | Mono-based | Mono | SRC | 97.72 | 83.08 |
| | | Mono-HOG | SVM | 98.15 | 58.65 |
| | | Mono-BoVW | SVM | 98.02 | 19.11 |
| | | Mono-Grass | SRC | 98.61 | 24.23 |
| | Pol-based | Polarimetric decomposition | SVM | 98.30 | 32.65 |
| | | Polarimetric scattering coding | SVM | 97.65 | 39.94 |
| | others | Steerable Wavelet | SVM | 98.89 | 38.54 |
| | | ASC | SVM | 98.46 | 15.49 |
| Deep features | CNN-based | A-ConvNet | Softmax | 97.99 | 442 |
| | | CV-CNN | Softmax | 98.46 | 403.91 |
| | | CV-FCNN | Softmax | 98.98 | 341.89 |
| | | CVNLNet | Softmax | 99.44 | 320.46 |
| | | RVNLNet | Softmax | 98.52 | 431.71 |
| | Transformer-based | ViT | Softmax | 98.77 | 389.2 |
| | | SpectralFormer | Softmax | 98.12 | 376.27 |
| | | CrossViT | Softmax | 99.17 | 363.07 |
| | others | SymNet | KNN | 97.28 | 263.28 |
| | | Monogenic ConvNet layer | Softmax | 98.73 | 308.70 |
| Multi-features | FEC | | Softmax | 99.10 | 195.06 |
| | Mono-CVNLNet | | Softmax | 99.54 | 227.96 |
| | **Proposed** | | Softmax | **99.75** | 322.93 |

Table 4 indicates that the accuracy rate achieved by the proposed MF-DCMANet is the highest, with a value of 99.75%. In addition, from Figure 9a, the values of the diagonal elements are very close to 360, and only eight samples in total are misclassified in the off-diagonal elements. The proposed method MF-DCMANet significantly outperforms the single feature-based methods and the multi-feature-based methods, and this can mainly be attributed to the proposed cross-fusion method effectively integrates the classification information contained in different features through the CFN and CMA module to obtain more abstract high-level semantic features. Compared to MF-DCMANet, the handcrafted Mono-Grass and Polarimetric decomposition models are less robust, although they are the best-performing methods in the Mono-based and Pol-based approaches, respectively, the proposed method resulted in an improvement of 1.14% and 1.39% in classification accuracy. From the perspective of classification accuracy, the effective utilization of multiple features leads to a greater improvement in performance, which also shows the necessity of multi-feature extraction. The last column of Table 4 displays the FPS (Frames Per Second) values comparison among different methods. FPS refers to the number of images that a method can process within one second. It can be seen that the FPS values based on deep learning methods are much higher than those of handcrafted feature methods. Moreover, the proposed method achieves an FPS value of 322.93, which is higher than other multi-feature fusion methods and roughly equivalent to the FPS values of methods based on CNN or Transformer.

4.4.2. Classification Accuracy Evaluation under Few-Shot Recognition

In few-shot recognition scenarios, we selected the methods with the highest accuracy rate under each small category in Table 4 for comparison. These methods are Mono-Grass,

Polarimetric decomposition, Steerable Wavelet, CVNLNet, CrossViT, Monogenic ConvNet layer, FEC, and Mono-CVNLNet, respectively. The dataset used for few-shot recognition was constructed by sampling the original GOTCHA dataset at ratios of 1/3, 1/5, 1/7, and 1/10. In Figure 10, it is apparent that regardless of the variation in the number of training samples, the accuracy curve of MF-DCMANet consistently remains at the top, indicating greater recognition performance than other approaches. Compared with the best-performing Mono-CVNLNet [10] in previous research works, our approach shows a significant improvement in few-shot recognition. Specifically, in 1/5 and 1/7 few-shot recognition scenarios, the proposed method resulted in an improvement of approximately 2% in performance, indicating that the final obtained features can more effectively capture the latent information of PolSAR targets and are insensitive to azimuth changes. Furthermore, we can observe from Figure 10 that the methods based on feature fusion, such as FEC, Mono-CVNLNet, and proposed MF-DCMANet, can maintain relatively high recognition accuracy even as the training sample size is rapidly reduced. In contrast, the methods that rely on a single feature, such as Wavelet and Polarimetric decomposition, experience sharp drops in the recognition accuracy when the training sample size is reduced to 1/7 and 1/10. Comparing the performance of Mono-CVNLNet and CVNLNet methods, it becomes apparent that both have similar recognition accuracy when the size of the training sample is adequate. As the amount of training data is reduced to 1/5 and 1/7, Mono-CVNLNet always performs better than CVNLNet, and the recognition accuracy is improved by about 3% and 6%, respectively. This is because, compared with deep features, handcrafted features can describe PolSAR targets robustly and stably without being constrained by the number of samples, which is why we have to introduce manually designed polarization and monogenic features in the proposed MF-DCMANet.

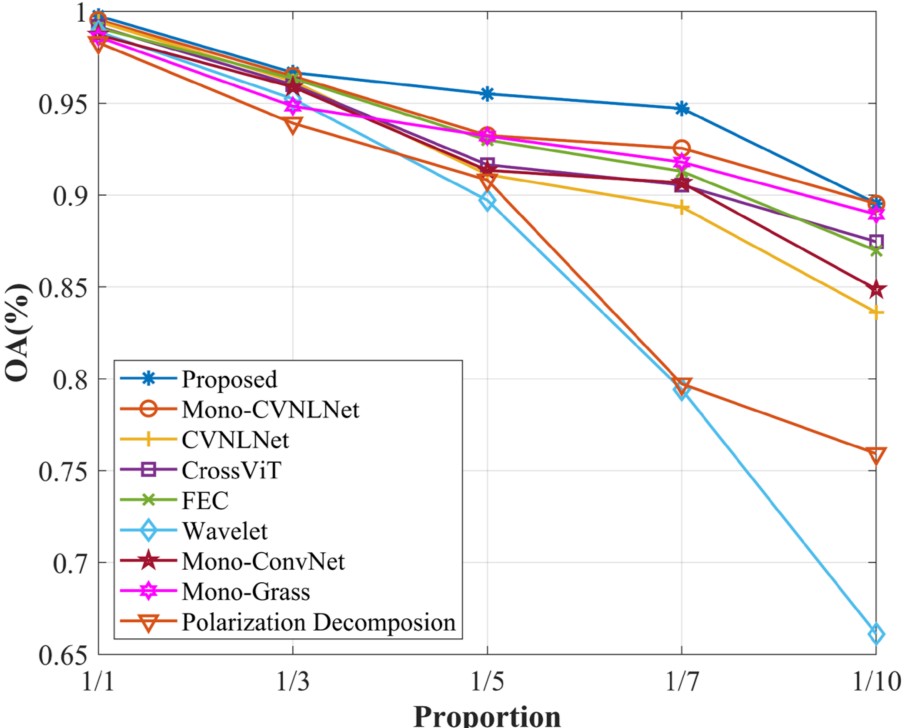

**Figure 10.** Accuracy of different methods in few-shot recognition.

In Figure 11, we graphed the ROC curves of the different methods for 1/10 few-shot recognition on the same coordinate system to facilitate a straightforward comparison of their performance. The results show that the ROC curve of the proposed method is closest to the upper left corner, and there is no intersection with other comparison methods, and it completely envelopes the ROC curves of other methods. Correspondingly, its AUC value reaches the highest at 0.98104, indicating that the proposed method has excellent generalization ability.

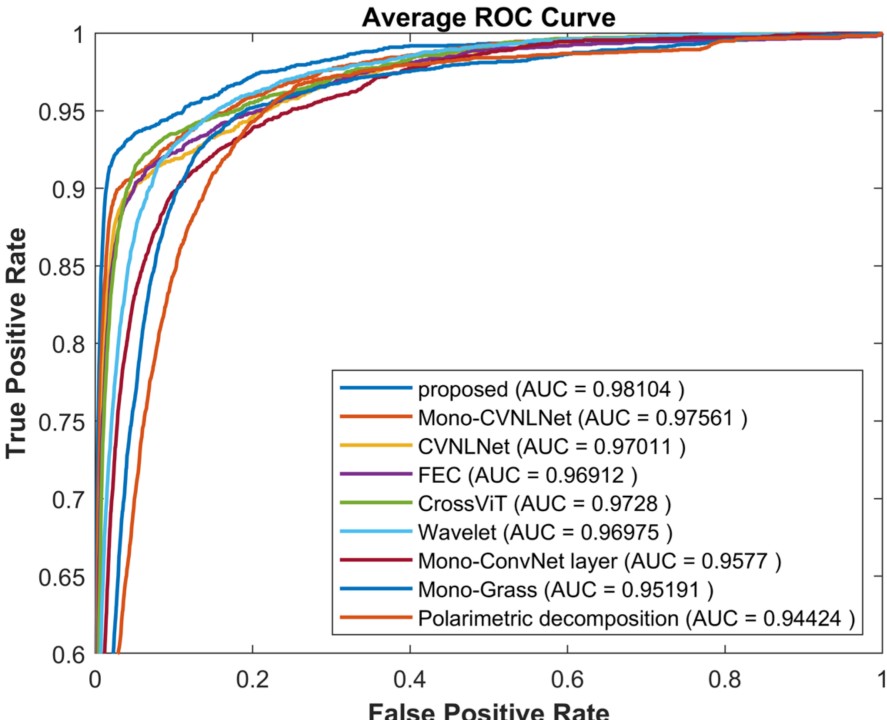

**Figure 11.** ROC curves and AUC values of different methods in the 1/10 few-shot recognition experiment.

4.4.3. Classification Accuracy Evaluation under Open Set Recognition

In open set recognition, the set of training categories is included in the set of test categories, which means that the test samples could belong to either the trained categories or the untrained categories [66]. We have developed two methods for identifying unknown categories in the open-set recognition scenario, where categories 2 and 3 were omitted from the training set but included in the test set. The first method involves setting a threshold on the output of the softmax, and test samples with values smaller than the threshold are judged as unknown categories. The second method is to extract features obtained after two-stage cross-fusion, i.e., the output of the CMA module. Then, we computed the feature centers of all known categories in the training set using the k-means algorithm. During the testing stage, the KL divergence between the test sample and the feature centers of each known category is computed. The category of the testing sample is determined by setting a threshold. The overall accuracy of the two methods at different thresholds is shown in Figure 12. As seen in Figure 12a, the first method achieves the highest overall accuracy of 86.45% when the threshold value of softmax is set to 0.75. However, when the threshold is exceeded, the overall accuracy decreases rapidly, indicating that the network gives false predictions with high confidence even in the case of unknown PolSAR targets. In Figure 12b, the highest overall accuracy of 91.23%, an increase of 4.78% compared to the first method, is achieved when the KL divergence threshold is set to 2.3. This is because the proposed method can effectively reduce intra-class distance while increasing inter-class distance, and KL divergence, as a distance metric, can measure the subtle distribution differences between different categories. Specifically, the average KL divergence between different categories is shown in Figure 13. The seven dark blue diagonal values in Figure 13

represent the average KL divergence between the test samples of known categories and the seven feature centers calculated during training. The eighth and ninth rows in Figure 13 record the average KL divergence between the test samples of unknown categories and the seven feature centers. It can be observed that the average KL divergence between samples of the same category is much smaller than that between different categories. Therefore, it is easy to set a threshold to quickly reject unknown samples in the test set. In this work, the threshold is set to 2.3.

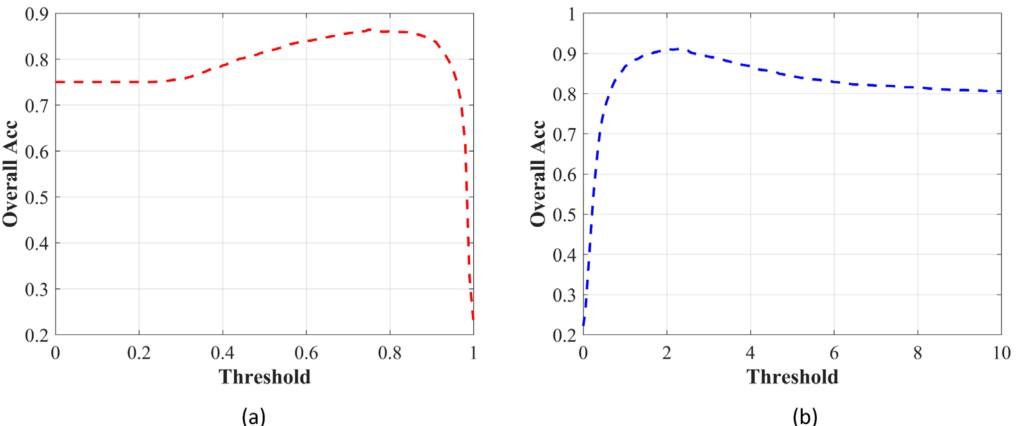

(a)

(b)

**Figure 12.** The overall accuracy of different methods at different thresholds. (**a**) Softmax (**b**) KL divergence.

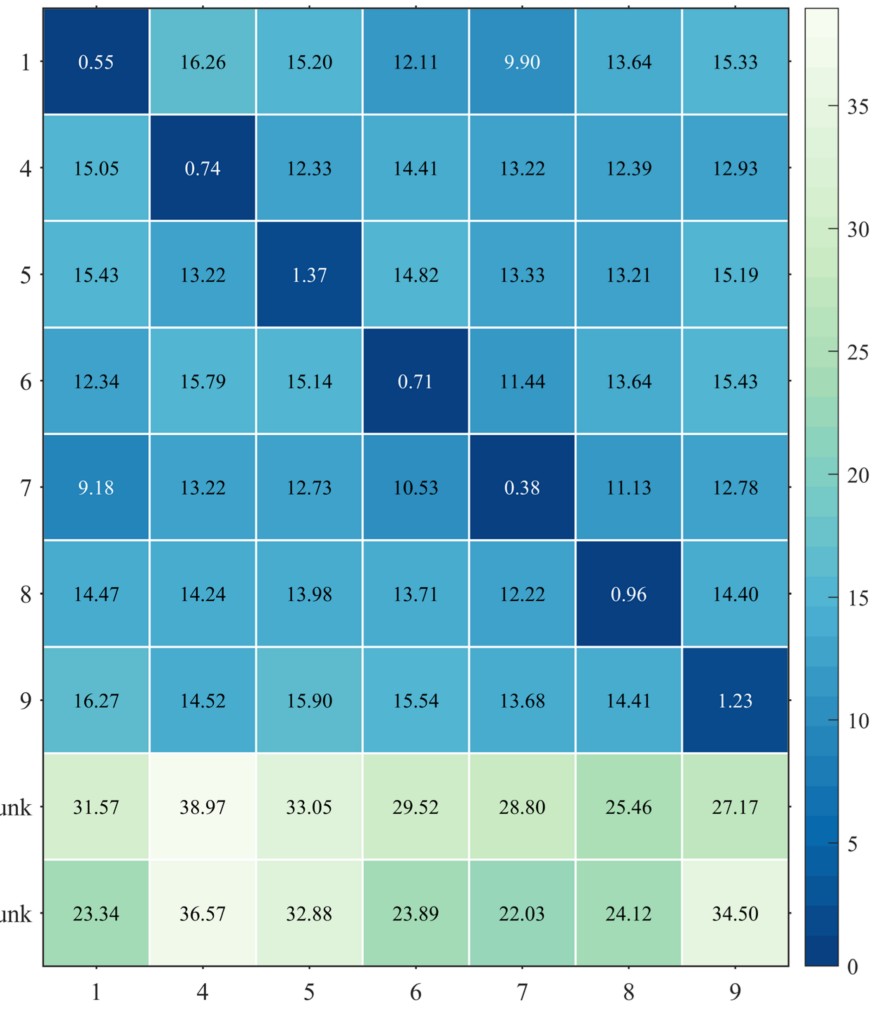

**Figure 13.** The average KL divergence between different categories.

To demonstrate the impact of two-stage cross-fusion in the proposed method, we extract the output features of both the CFN and CMA modules for calculating the feature centers required by the KL-divergence method. When the threshold of KL divergence is set to 2.3, the corresponding confusion matrices are shown in Figure 14. Notice that the last two rows of the confusion matrix represent the prediction results of unknown categories in the test set, while the last column represents the situation where all categories of the test set are predicted as unknown categories. The results indicate that the classification performance after two-stage cross-fusion is superior to that after one-stage cross-fusion, with an overall accuracy improvement of 6.23%.

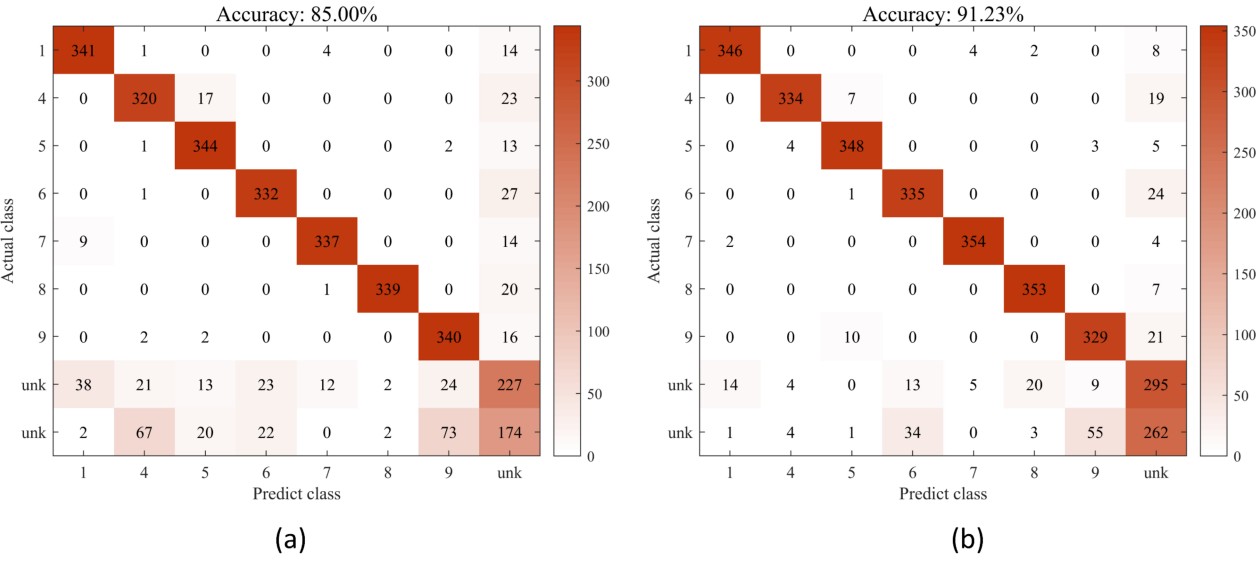

(a) (b)

**Figure 14.** Confusion matrix of the proposed method at different feature fusion stages. (**a**) The first stage: CFN module; (**b**) the second stage: CMA module.

To reflect the superiority of the proposed method, we apply the method used for comparison in Section 4.4.2 to open set recognition, and the experimental results are shown in Table 5. The proposed MF-DCMANet achieved known target accuracy, unknown target accuracy, and overall target accuracy of 95.20%, 77.36%, and 91.23%, respectively. Compared to the second-best method, Mono-CVNLNet, the proposed method improved the performance by 1.39%, 2.22%, and 1.57%, respectively. In Table 5, known target accuracy is defined as the ratio of correctly classified known targets to the total number of known targets in the test set. Unknown target accuracy represents the ratio of correctly classified unknown targets to the total number of unknown targets in the test set. The overall accuracy rate represents the ratio of correctly classified known and unknown targets to the total number of targets in the test set.

**Table 5.** Accuracy comparison of different methods under open-set recognition experiments.

| Method | Known Target Accuracy (%) | Unknown Target Accuracy (%) | Overall Target Accuracy (%) |
|---|---|---|---|
| Mono-Grass | 89.96 | 67.64 | 85.00 |
| Mono-ConvNet | 88.89 | 72.78 | 85.31 |
| Wavelet | 92.30 | 70.00 | 87.35 |
| CrossViT | 92.78 | 71.11 | 87.96 |
| CVNLNet | 93.97 | 69.58 | 88.55 |
| FEC | 93.29 | 73.19 | 88.83 |
| Mono-CVNLNet | 93.81 | 75.14 | 89.66 |
| **Proposed** | **95.20** | **77.36** | **91.23** |

4.4.4. Ablation Study

This subsection serves to validate the performance of three critical modules in the MF-DCMANet: Multi-Feature (MF) extraction, Cross-Feature Network (CFN), and Cross-Manifold Attention (CMA) transformer.

We evaluated the capability of the framework with only monogenic features or polarization features as inputs to the MF module. At this time, we noted that the cross-attention mechanism in the CMA module degenerates into a self-attention mechanism with a local window. The framework with single-feature input (another feature is zeroed) is named M-DSMANet and P-DSMANet. As shown in Table 6, compared to the framework with single-feature input, the integration of two handcrafted features into the framework increases the amount of information available, resulting in higher accuracy as evidenced by the classification accuracy increased by 1.17% and 1.91%.

**Table 6.** Ablation experiment of multi-feature extraction module.

| Method | M-DSMANet | P-DSMANet | MF-DCMANet |
|---|---|---|---|
| OA (%) | 98.58 | 97.84 | 99.75 |

To explore the functionality of the CFN module in the MF-DCMANet, we selected three other common fusion methods as comparison methods: Concat, Parallel (add), and En-De [67], which directly replace the cross-fusion process in the CFN module.

**Concat**: Directly concatenate the mid-level monogenic features $M_d$ and the mid-level polarization features $P_d$ along the channel dimension.

**Parallel**: Directly add the mid-level monogenic features $M_d$ and the mid-level polarization features $P_d$ along the channel dimension.

**En-De**: Perform encoder-decoder fusion on $M_d$ and $P_d$.

Table 7 shows that the cross-fusion method outperforms these three comparative fusion methods in terms of recognition accuracy, with accuracy rates increasing by 1.20%, 1.02%, and 4.07%, respectively. This indicates that the CFN module can effectively and globally interact with the information carried by different features compared to the simple feature fusion method. In the cross-fusion process, the information carried by the polarization feature and the monogenic feature is continuously updated interactively during the backpropagation process of the network, resulting in a feature representation that captures the invariant in the PolSAR targets. As a result, better recognition performance can be achieved.

**Table 7.** Ablation experiment of Cross-Feature Network (CFN) module.

| Method | Concat | Parallel | En-De | CFN |
|---|---|---|---|---|
| OA (%) | 98.55 | 98.73 | 95.68 | 99.75 |

In the ablation experiment of the CMA module, we return the calculation of attention weight *A* to Euclidean space, whereby we calculate the cosine similarity of feature vectors instead of calculating the Grassmann measure of feature patches. As shown in Table 8, computing attention weight directly in the Euclidean space reduces the final recognition accuracy by 1.11%. In contrast to the Euclidean space, the Grassmann manifold space reflects the high-dimensional geometric relationships of the data and provides a more detailed description of the similarity between feature patches through the projection distance. Therefore, the CMA module addition assists the MF-DCMANet in achieving better performance.

**Table 8.** Ablation experiment of Cross-Manifold Attention (CMA) module.

| Method | Euclidean | Grassmann |
|--------|-----------|-----------|
| **OA (%)** | 98.64 | **99.75** |

### 4.5. Qualitative Analysis

#### 4.5.1. CFN Module Analysis

The representation performance of a single feature is largely limited due to the lack of discriminative nature of the extracted features, particularly in few-shot recognition scenarios. In this paper, the monogenic features utilize the multi-scale characteristics to describe the scattering phenomenon of the PolSAR target, while the polarimetric features extracted from the polarization covariance matrix capture all target polarization information obtained from the radar measurement, which finely describes the target scattering mechanism from the physical meaning. The proposed CFN module effectively utilizes the complementary advantages between these features to enhance the representation ability of a single feature.

To visualize the original monogenic features, polarization features, and fused features obtained by the CFN module in 2D space, we utilized the t-SNE method [68]. This method is known for its ability to reduce high-dimensional features to 2D space, expand dense clusters, and shrink sparse clusters for optimal visualization. Figure 15 presents the visualized features, which illustrate that the adequately trained CFN module can extract informative features that fully absorb the unique classification information carried by the polarization and monogenic features. Additionally, the CFN module can easily separate different types of targets by making the clusters from the same category more compact and those from different categories more dispersed.

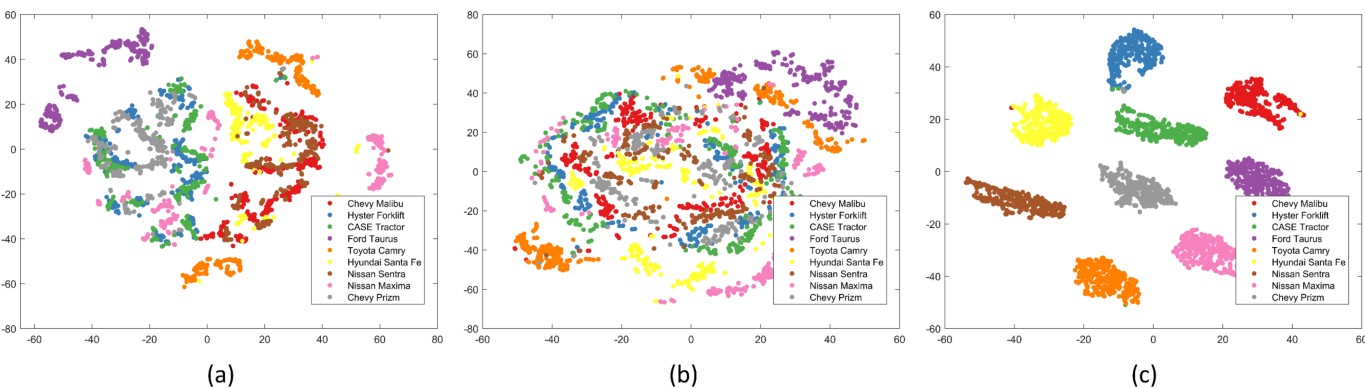

**Figure 15.** Visualization features of GOTCHA data by using t-SNE. (**a**) the original monogenic features. (**b**) the original polarization features. (**c**) the fused features obtained by the CFN module.

#### 4.5.2. CMA Module Analysis

In Section 3.4, we mentioned that the attention weight of the self-attention mechanism actually computes the cosine similarity between feature vectors in the Euclidean space, while the CMA module calculates the projection distance between feature patches in the Grassmann manifold space, which can be associated with the principal angle. We extracted the query $Q \in \mathbb{R}^{L \times D}$ and key $K \in \mathbb{R}^{L \times D}$ of a PolSAR target, where $L = 9, D = 128$. We calculated the similarity between the 128-dimensional feature vectors in Euclidean space and converted them into angle values, as shown in Figure 16. It can be observed that the cosine angle between most of the feature vectors is around $90°$, indicating that the vectors are in an orthogonal relationship. Their product is either 0 or a small value, and the information carried by the feature vectors is offset. This is mainly because the monogenic features and polarization features describe the PolSAR target from different perspectives, which results in an insufficient correlation between the vectors from different features.

This is evidenced by their orthogonality in the Euclidean space, making it difficult for subsequent feature extraction to create meaningful feature representations.

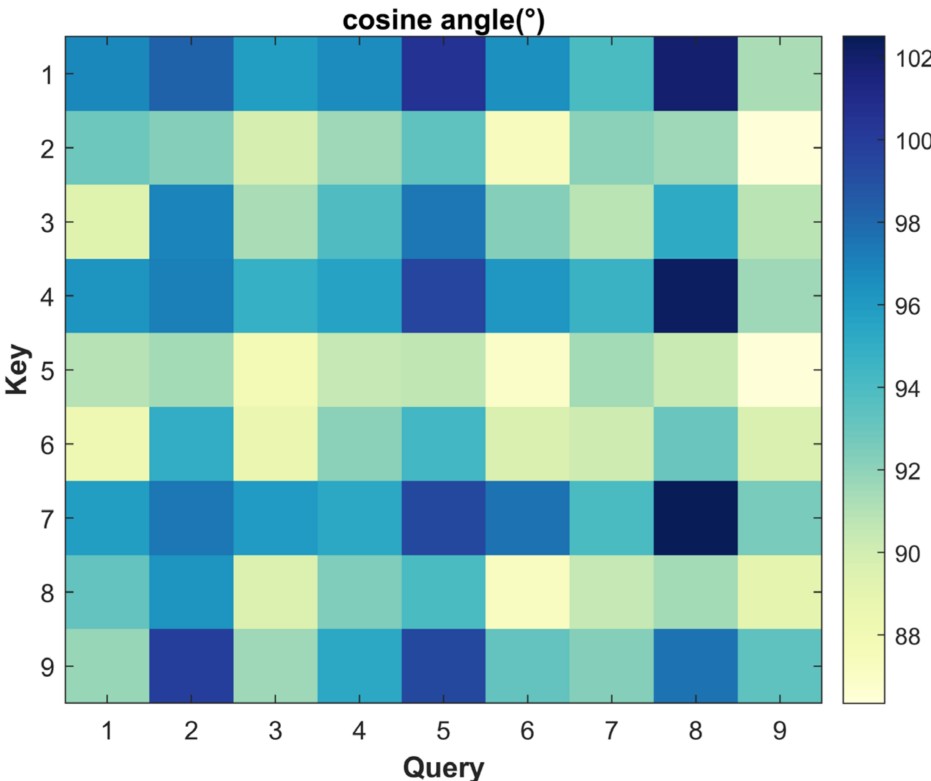

**Figure 16.** The cosine angles between query vectors and key vectors.

Next, we convert the query $Q \in \mathbb{R}^{L \times D}$ and key $K \in \mathbb{R}^{L \times D}$ to query $Q \in \mathbb{R}^{L \times n \times k}$ and key $K \in \mathbb{R}^{L \times n \times k}$, and represent the feature patches $Q_i$, $K_j \in \mathbb{R}^{n \times k}$ on the Grassmann manifold, where $n = 16, k = 8$. Then we calculate the principal angle of different feature patches. In Figure 17, $span(Q_i)$ and $span(K_j)$ are formed by representing the query $Q_i$ and key $K_j$ in the Grassmann manifold space, where $span(Q_i)$ and $span(K_j)$ denotes the subspace spanned by columns of $Q_i$, $K_j \in \mathcal{G}(n, k)$. The principal angle between $span(Q_i)$ and $span(K_j)$ falls within the range of $[0°, 90°]$, which ensures that the vector product of the feature subspace does not produce a large number of zeros and enhances the correlation of different feature vectors. Therefore, the CMA module restricts the angles between different feature vectors to the range of $[0°, 90°]$ range by representing features on the Grassmann manifold, effectively avoiding the limitations of computing attention weight in Euclidean space. As such, it is more suitable for multi-feature-based PolSAR target recognition tasks and can obtain more valuable information to improve the recognition rate.

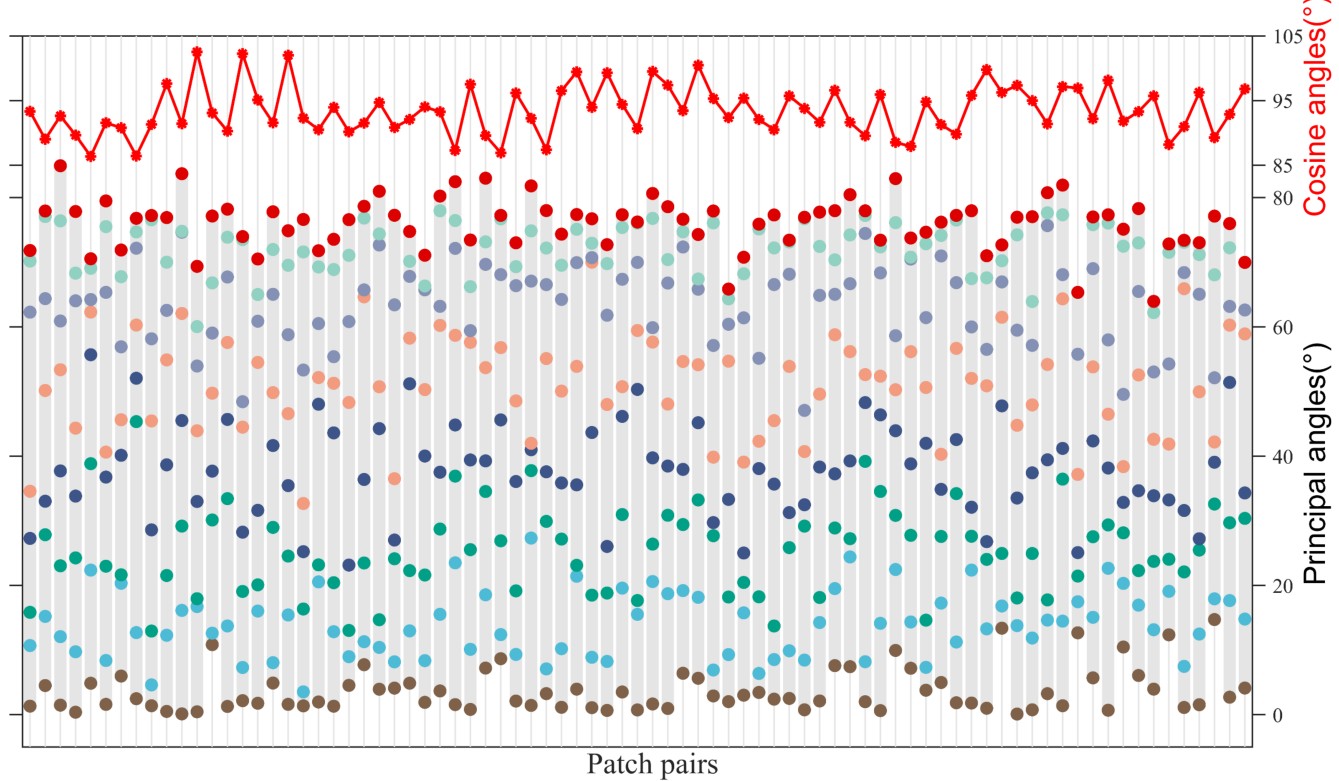

**Figure 17.** Comparison of Euclidean spaces and Grassmann manifolds. On the Grassmann manifold, the dimension of $Q$ and $K$ changes from $\mathbb{R}^{L \times D}$ to $\mathbb{R}^{L \times n \times k}$, where $L$=9. Thus, the horizontal axis of Figure 17 represents the 81 measures between $Q_i(i = 1 : 9)$ and $K_j(j = 1 : 9)$, while the upper and lower parts of the vertical axis represent the cosine angles in Euclidean space and principal angles in the Grassmann manifold space, respectively. We have plotted the 81 cosine angle values in Figure 16 as the top red polyline in Figure 17. According to the definition of principal angle in Equation (16), the number of principal angles between $Q_i$ and $K_j$ equals $k$ (where $k = 8$). Hence, the eight differently colored points within each gray stripe represent the eight principal angles of patch pairs.

## 5. Conclusions

We propose a novel multi-feature cross-fusion approach based on monogenic signal and polarization features for PolSAR target recognition, namely, MF-DCMANet. The proposed method employs a dual-stage cross-fusion process on the initially extracted monogenic features and polarization features, enabling features with different characteristics to continuously exchange information during network training, resulting in a feature representation with global and local attention. The proposed framework integrates feature extraction, fusion, and classification processes. The results on the constructed GOTCHA dataset indicate that taking into consideration the characteristics of the Grassmann manifold structure can effectively enhance the classification accuracy of PolSAR targets. Specifically, the proposed MF-DCMANet achieves the highest accuracy on the full dataset and maintains satisfactory overall performance in few-shot recognition and open-set recognition experiments, with a recognition accuracy improvement of approximately 2% compared to the current state-of-the-art method.

It should be noted that the fundamental mechanism of how the proposed method integrates multiple features and reduces the differences between them in deep neural networks has not been thoroughly analyzed. Moreover, while the effectiveness of the proposed method has been verified on vehicle targets, its applicability to other types of polarization targets remains to be explored. In the future, we plan to conduct a deeper investigation into the potential mechanism of deep neural network-based feature fusion and explore the possibility of applying the method to more types of polarization targets.

**Author Contributions:** Conceptualization, F.L. and X.Z.; methodology, F.L., C.Z. and X.Z.; software, C.Z.; validation, C.Z.; formal analysis, F.L., C.Z. and X.Z.; investigation, F.L. and X.Z.; resources, X.Z.; data curation, C.Z.; writing—original draft preparation, C.Z.; writing—review and editing, F.L. and X.Z.; visualization, C.Z.; supervision, F.L., Y.L. and X.Z.; project administration, F.L. and Y.L.; funding acquisition, F.L. and Y.L. All authors have read and agreed to the published version of the manuscript.

**Funding:** This research was funded by the National Key R&D Program of China (Grant No. 2018YFE0202102), China Postdoctoral Science Foundation (Grant No. 2021M690412), Natural Science Foundation of Chongqing, China (Grant No. cstc2020jcyj-msxmX0812).

**Data Availability Statement:** The GOTCHA dataset is available at https://www.sdms.afrl.af.mil/index.php?collection=gotcha, accessed on 14 January 2023.

**Conflicts of Interest:** The authors declare no conflict of interest.

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
