# Peer review of "MF-DCMANet: A Multi-Feature Dual-Stage Cross Manifold Attention Network for PolSAR Target Recognition"

_remotesensing, doi:10.3390/rs15092292_

Round 1

Reviewer 1 Report

The authors have proposed a framework for PolSAR target recognition, and the paper is well-structured overall. However, the quality of almost all images in the article is low, so the authors should consider using other formats, such as PDF if they used LaTeX. Figure 9 shows the confusion matrices of the method under different training schemes (full training datasets, 1/3, 1/7, and 1/10 of the training set). Instead of downsampling, it would be interesting if the authors omitted one or two classes from the training set but kept them in the test dataset. This approach would create an open-set recognition scenario, also known as out-of-distribution detection with a rejection option. The simplest way to recognize unknown classes would be to apply a threshold on softmax outputs. Finally, the confusion matrix can be modified to include one or more input unknown classes, in addition to the known input and output classes, while having only one output unknown class. For more information on open-set recognition in SAR images, we recommend the article proportional similarity-based openmax classifier for open set tecognition in SAR images.  Page 3: It is commendable that the authors have used bullet points to express the contributions of the paper. However, it would be better if each item was described concisely and directly to the point, expressing the contribution precisely.

Reviewer 2 Report

Journal: Remote Sensing

Manuscript ID: remotesensing-2348419

Manuscript Title: MF-DCMANet: A Multi-Feature Dual-stage Cross Manifold Attention Network for PolSAR Target Recognition

Author: Feng Li, Chaoqi Zhang, Xin Zhang and Yang Li

Good work. Please solve the following comments with a major revision and to be reviewed again.

1.        In Abstract, please give some specific accuracy indictors.

2.        The statement of contributions at the end of Sec. I should be enriched more, also aiming at better highlighting the relevant technical challenges tackled by the authors.

3.        When mentioning about the CNN-Based target recognition field, the authors should also briefly mention literature on SAR ship recognition, i.e., hog-shipclsnet: a novel deep learning network with hog feature fusion for sar ship classification, a polarization fusion network with geometric feature embedding for sar ship classification, squeeze-and-excitation laplacian pyramid network with dual-polarization feature fusion for ship classification in sar images, injection of traditional hand-crafted features into modern cnn-based models for sar ship classification: what, why, where, and how.

4.        Increase the readability of all figures.

5.        Give more information about the dataset.

6.        Since the data sources are limited (all from the same scene regions), I wonder if there exists the risk of overfit during the training.

7.        Add a section of Experimental Platform.

8.        In Sec. of Related Works, the authors should provide richer context on the successful application of dl to several sar application field, such as sar ship detection, sar ship segmentation and sar shadow tracking: balance learning for ship detection from synthetic aperture radar remote sensing imagery, a group-wise feature enhancement-and-fusion network with dual-polarization feature enrichment for sar ship detection, a mask attention interaction and scale enhancement network for sar ship instance segmentation, htc+ for sar ship instance segmentation, shadow-background-noise 3d spatial decomposition using sparse low-rank gaussian properties for video-sar moving target shadow enhancement, high-speed ship detection in sar images based on a grid convolutional neural network, a full-level context squeeze-and-excitation roi extractor for sar ship instance segmentation, hyperli-net: a hyper-light deep learning network for high-accurate and high-speed ship detection from synthetic aperture radar imagery.

9.        Add the test time of different methods.

10.     In Table 4, it is suggested to change the classifier of FEC and Mono-CVNLNet into Softmax, so as to keep the same as the proposed method in terms of the classifier.

11.     IMHO, the Conclusion should be re-written to 1) explicitly describe the essential features/advantages of the review that other reviews do not have, and 2) describe the limitation(s) of the review.

12.     The English should be improved greatly.

Minor editing of English language required

Round 2

Reviewer 2 Report

Accept in present form. All comments have been solved. Good work.

Minor editing of English language required.